# Towards development of a statistical framework to evaluate myotonic dystrophy type 1 mRNA biomarkers in the context of a clinical trial

**Adam Kurkiewicz**[1]*, **Anneli Cooper**[3¤], **Emily McIlwaine**[3], **Sarah A. Cumming**[3], **Berit Adam**[3], **Ralf Krahe**[4], **Jack Puymirat**[5], **Benedikt Schoser**[6], **Lubov Timchenko**[7], **Tetsuo Ashizawa**[8], **Charles A. Thornton**[9], **Simon Rogers**[2☯], **John D. McClure**[1☯], **Darren G. Monckton**[3☯]

1 Institute of Cardiovascular and Medical Sciences, University of Glasgow, Glasgow, United Kingdom, 2 School of Computing Science, University of Glasgow, Glasgow, United Kingdom, 3 Institute of Molecular Cell and Systems Biology, College of Medical, Veterinary and Life Sciences, University of Glasgow, Glasgow, United Kingdom, 4 Department of Genetics, University of Texas, MD Anderson Cancer Center, Houston, TX, United States of America, 5 Laboratory of Human Genetics, CHUL Medical Research Centre, University of Laval, Quebec City, QC, Canada, 6 Department of Neurology, Friedrich Baur Institute, Ludwig Maximilians University, Munich, Germany, 7 Department of Pediatrics, Division of Neurology, Cincinnati Children's Hosptial, University of Cincinnati, College of Medicine, Cincinnati, Ohio, United States of America, 8 VA Medical Center, Houston, Texas, United States of America, 9 University of Rochester, Medical Center School of Medicine and Dentistry, Rochester, New York, United States of America

☯ These authors contributed equally to this work.
¤ Current address: Institute of Biodiversity, Animal Health and Comparative Medicine, University of Glasgow, Glasgow, United Kingdom
* Adam.Kurkiewicz@glasgow.ac.uk

**Data Availability Statement:** All data has been submitted to Array Express and is publicly available as E-MTAB-7983: https://www.ebi.ac.uk/arrayexpress/experiments/E-MTAB-7983/.

## Abstract

Myotonic dystrophy type 1 (DM1) is a rare genetic disorder, characterised by muscular dystrophy, myotonia, and other symptoms. DM1 is caused by the expansion of a CTG repeat in the 3'-untranslated region of *DMPK*. Longer CTG expansions are associated with greater symptom severity and earlier age at onset. The primary mechanism of pathogenesis is thought to be mediated by a gain of function of the CUG-containing RNA, that leads to *trans*-dysregulation of RNA metabolism of many other genes. Specifically, the alternative splicing (AS) and alternative polyadenylation (APA) of many genes is known to be disrupted. In the context of clinical trials of emerging DM1 treatments, it is important to be able to objectively quantify treatment efficacy at the level of molecular biomarkers. We show how previously described candidate mRNA biomarkers can be used to model an effective reduction in CTG length, using modern high-dimensional statistics (machine learning), and a blood and muscle mRNA microarray dataset. We show how this model could be used to detect treatment effects in the context of a clinical trial.

**Funding:** Sample collection and initial data analysis were funded by the Marigold Foundation in the form of international collaboration "Dystrophia Myotonica Biomarker Discovery Initiative". The sponsors played active role in the study design and data collection, by liaising with clinicians contributing muscle and blood samples, making arrangements with the company, which carried out the microarray work, and carrying out the initial data analysis. The sponsors played no role in preparation of the manuscript or a decision to publish.

**Competing interests:** Adam Kurkiewicz Declares ownership of Illumina and PacBio shares. Anneli Cooper Has served on a Scientific Advisory Board for AstraZeneca (Trypanosomiasis). Sarah Cumming None declared Berit Adam None declared Ralf Krahe None declared. Jack Puymirat None declared Benedikt Schoser Benedikt Schoser is member of the Neuromuscular advisory board of Audentes Therapeutics, USA, and Scientific advisory of Nexien BioPharm, USA. He received speaker honoraria from Sanofi Genzyme, Amicus Therapeutics, Lupin Pharmaceuticals, and Kedrion. He received an unrestricted research grant from Sanofi Genzyme USA (2016-2019), Greenovation FRG (2017-2020), and from the Marigold foundation (2014) Lubov Timchenko None declared. Tetsuo Ashizawa 1. My wife has stocks and stock options of BIOPATH Holdings, Inc. 2. One US and international patent approved. 3. Grants from NIH, the Myotonic Dystrophy Foundation, the National Ataxia Foundation and Biogen. 4. I am a member of the advisory board for the National Ataxia Foundation and that for the Myotonic Dystrophy Foundation. Simon Rogers None declared John McClure None declared Darren G Monckton Professor Monckton has been a scientific consultant and/or received an honoraria or stock options from Biogen Idec, AMO Pharma, Charles River, Vertex Pharmaceuticals, Triplet Therapeutics, LoQus23, BridgeBio, Small Molecule RNA and Lion Therapeutics. Professor Monckton also had a research contract with AMO Pharma and CHDI has received research grants from the European Union, European Huntington Disease Network, National Institute of Health, Muscular Dystrophy UK and the Myotonic Dystrophy Support Group. Professor Monckton is on the Scientific Advisory Board of the Myotonic Dystrophy Foundation, is a scientific advisor to the Myotonic Dystrophy Support Group and is a vice president of Muscular Dystrophy UK. Charles Thornton Dr. Thornton has received sponsored research support from Ionis Pharmaceuticals, Biogen, Genzyme, and Dyne Therapeutics, and research grants from the National Institutes of

# Introduction

## Myotonic dystrophy type 1

Myotonic dystrophy type 1 (DM1) is an autosomal dominant trinucleotide repeat disorder, caused by an expanded CTG repeat in the 3' UTR of the *dystrophia myotonica* protein kinase (*DMPK*) gene [1]. Transcription of *DMPK* in affected individuals produces a toxic, GC-rich mRNA molecule, which results in dysregulation of several RNA binding factors, including proteins MBNL1, MBNL2, MBNL3, CELF1 (CUGBP1), HNRNPH1 and STAU1 (Staufen1) [2, 3]. The pathomechanism of the dysregulation of splicing factor MBNL1 is perhaps best understood, with MBNL1 sequestration to the toxic *DMPK* RNA product resulting in alternative splicing defects of pre-mRNAs of multiple genes, including the chloride channel (*CLCN1*), brain microtubule-associated tau (*MAPT*) and insulin receptor (*INSR*) [4]. Such alternative splicing (AS) defects are generally believed to be a major contributing factor of clinical symptoms of DM1, such as myotonia (*CLCN1*) or abnormal glucose response (*INSR*), and have been postulated to play a role in cardiac conduction defects (*RYR2*, *SERCA2*, *TNNT2*) [5]. AS defects may underlie other clinical symptoms of DM1, including muscle wasting, cataracts, hypersomnia, gastrointestinal abnormalities, as well as premature baldness and testicular atrophy in males [2, 6]. The severity of symptoms is positively correlated with CTG repeat length [2].

The toxic *DMPK* transcript in DM1-affected individuals has been identified as an active target for theraupetic intervention [7], and it is expected that breakdown of the toxic mRNA will result in at least partial reversal of DM1-induced AS changes and other known and unknown DM1-induced biomolecular pathologies.

Spliceopathy of DM1 is an active area of research, with novel splicing defects being continuously reported. Nakamori *et al.* [8] identified a set of 41 genes, which are mis-spliced in DM1, suggesting these genes as potential biomarkers of DM1. Batra *et al.* [9] identified 80 genes, whose expression indicates disrupted APA, reusing the (human) dataset of Nakamori *et al.* and using other datasets, including data from mouse models.

## Predictors in genetics research

A widespread paradigm in biological and clinical research is the case-control study, using frequentist statistics tools focusing on hypothesis testing (inference). Examples of such designs include Genome Wide Association Studies (GWAS) [10], placebo and active control clinical trial designs [11], non-inferiority designs [12] or heredity designs based on twin studies. It is reported that designs of as many as 70% of studies published in leading medical journals use at most the following three statistical tests as part of their design: Student's t-test, Fisher's exact test, and the Chi-square test [13].

A possible alternative to the focus on hypothesis testing is building predictors or classifiers, which produce a numerical estimate of a given trait (height, size of the DM1 trinucleotide expansion) or effect size, predict participant's category (such as affected/unaffected), or estimate the effect size of the treatment, given a set of independent variables (e.g. genotypes, mRNA profiles, *etc.*). If necessary, the efficacy of these predictors/classifiers can be evaluated using traditional frequentist tools, such as *p*-values.

Predictors have been successfully applied in genetics research. For example, Lello *et al.* [14] report predicting human height from genotype data, obtained using human Single Nucleotide Polymorphism (SNP) microarrays, to within a few centimeters for most participants in their sample. This level of accuracy is achieved due to a very large sample size of nearly half a million individuals. A review by Van Raden *et al.* [15] reports ability to predict dairy output of certain cattle breeds with $R^2$ of 49%, using non-linear models based on SNP microarrays. The work of

Health, Food and Drug Administration, Muscular
Dystrophy Association, and Myotonic Dystrophy
Foundation. Dr Thornton has served on the
Scientific Advisory Board for Dyne Therapeutics.
This does not alter our adherence to PLOS ONE
policies on sharing data and materials.

Azencott *et al.* [16] allows one to incorporate prior information about biological networks into the predictive model, increasing prediction accuracy. In one of the direct motivations behind our research here, Lee *et al.* [17] report being able to predict most of Huntington disease trinucleotide repeat size using mRNA profiling of lymphoblastoid cell lines. Here, we demonstrate a further application of predictors in genetics research, by constructing a predictor, which can produce a numerical estimate of a participant's DM1 CTG repeat length (measured from blood) from an mRNA profile (obtained from muscle), and demonstrate its usefulness in the context of a hypothetical clinical trial of a DM1 treatment.

## Prior identification of alternative splicing and alternative polyadenylation events in muscles of DM1-affected individuals

Our primary reference is the work of Nakamori *et al.* [8], who identified 42 genes exhibiting AS defects in DM1. Briefly, the methodology of the study was as follows: muscle tissue (patients: four biceps, two quadriceps, one tibialis anterior, one diaphragm; controls: eight vastus lateralis) were sampled post-mortem from eight DM1 affected individuals and eight unaffected controls. Selection of the postmortem DM1 samples were based on high integrity of RNA present in the sample and presence of splicing misregulation of *INSR* and *AP2A1*, and then compared to quadriceps biopsy samples from eight unaffected controls.

mRNA was extracted from the samples, purified and hybridized to GeneChip™ Human Exon 1.0 ST microarrays. Putative AS defects were identified using a mixture of existing methods, such as Affymetrix's PLIER, DABG and Alternative Transcript Analysis Methods for Exon Arrays, and new methods proposed and described by the authors. Identified putative defects were validated using RT-PCR in 50 DM1 subjects, yielding 42 genes with confirmed AS defects.

Nakamori *et al.* report several technical obstacles with this approach, with the initial version of their pipeline suffering from as many as 80% putative AS defects failing to replicate with Reverse Transcription-PCR. Further analysis suggested that this occurred when signal intensity for the entire transcript or a particular exon was low, overall expression of a transcript was strongly up- or down- regulated in DM1 relative to normal controls, or signal intensity of an exon was inappropriately high relative to other exons in the same transcript [8].

Batra *et al.* [9] re-used the dataset of Nakamori *et al.* [8] and used similar techniques to filter down the data, in a search for genes with dysregulated APA. Their selection criteria focused on probesets with over 2-fold change in DM1 or DM2 vs. unaffected controls, excluding genes that were represented by $\leq 5$ probesets, retrogenes and non-protein coding genes, which resulted in pre-selection of 438 probesets. The authors performed visual inspection of all pre-selected probesets identifying 123 APA events belonging to 80 genes.

## Evaluating potential biomarkers of DM1

It is well established that the primary determinant of both age at onset and many progressive DM1 symptoms is the CTG repeat length [18–20]. Separate efforts have shown that case/control status in DM1 results in detectable AS changes in the mRNA profile of skeletal muscle (Nakamori et al. 2013; Batra et al. 2014). Our work acts as a bridge unifying both lines of investigation. In this research, rather than regarding DM1 status as a binary variable, we view DM1 as a spectrum of disease, the severity of which is quantified by the length of the DM1 CTG repeat in any individual patient. Based on the knowledge of the causative effect of CTG expansion on downstream pathology, and the important role of AS defects in this pathology, we thought it should be possible to capture the effect, which the length of the CTG repeat has on mRNA expression in muscle into a simple statistical model, based on partial least squares

regression (PLSR). Using the model we can predict the size of the DM1 CTG repeat from the mRNA profile significantly better than a random predictor. We propose that the model can serve as a valuable tool in evaluating efficacy of any treatment for DM1 as such treatment enters pre-clinical or clinical trials, by enabling investigators to directly quantify the treatment effect as measured by effective reduction of DM1 CTG repeat length rather than simply assessing the shift in AS of any one transcript. As disease severity is directly tied to CTG length, using this approach has the potential to yield more clinically meaningful interpretations of AS changes. For instance, an AS event that is dramatically shifted to the opposite extreme by even a small increase in the number of CTG repeats will yield a large signal on a case-control basis, but may not be closely tied to disease severity within the DM1 population and thus may not make a good reporter of an intervention as it may require a unrealistically large effect size to revert it to the non-disease associated range. However, as such an AS event yields little discriminatory power with regard to CTG repeat within the DM1 population, it would not be selected as a major contributor to our predictive model. Thus, our model has the potential to enrich for AS events that have more direct clinical relevance rather than absolute changes.

We would like to stress here that "effective reduction" in the case of most candidate treatments will not be an actual reduction in the repeat length (with notable exception of candidate treatments based on gene editing). Rather, an "effective reduction" is a reduction of "effective repeat length", i.e. repeat length as judged by the degree of splicing changes. This can be demonstrated with an example of a patient with a certain pre-treatment repeat length, and both physiological and molecular symptoms characteristic for that repeat length. If such patient were to undergo an effective treatment, we would expect these symptoms to be partially reversed, and our predictive framework to predict a shorter repeat length than the patient's actual repeat length. Subject to the correctness of our understanding of the molecular pathophysiology of the splicing changes, which we rely on to predict the effective repeat length, we expect that the reversal of the splicing changes would occur immediately after the release of inactivated AS and APA factors, such as MBNL1. This release should in turn happen immediately after a candidate therapeutic were to reach a clinically significant level in the relevant tissue. We expect this to happen on a timescale of days to weeks, unlike the reversal of physiological symptoms, which we would expect to happen on longer timescales.

## Materials and methods

### The dataset

As part of the *Dystrophia Myotonica* Biomarker Discovery Initiative (DMBDI) microarray analysis was used to investigate mRNA profiles in quadriceps muscle from 36 participants, including 32 DM1 cases and four unaffected controls. One DM1 patient refused blood donation, thus genetic analyses were only performed on 35 participants. All DM1 cases genotyped were heterozygous for the expanded CTG repeat and the mode of the length of the DM1 CTG expansion (Modal Allele Length, MAL) was determined by small-pool PCR of blood DNA for 35/36 patients [21]. For this work we did not attempt to measure the repeat length from muscle, due to a very high degree of repeat instability in muscle cells [3] and associated difficulties in its experimental measurement. One patient refused blood donation. For each of the 35 blood-donating patients mRNA expression profiling of blood was performed using the Affymetrix GeneChip™ Human Exon 1.0 ST microarray. For 28 of 36 patients a successful quadriceps muscle biopsy was obtained. The muscle tissue was mRNA profiled using the same type of microarray. In total, a complete set of samples (blood and muscle) was obtained for 27 of 36 patients. mRNA profiling was carried out by the GeneLogic service lab (on a fee-for-service basis) using standard Affymetrix hybridisation protocol.

Principal Component Analysis (PCA) was performed on both blood and muscle profiles, and visual inspection was carried out to detect the presence of outliers. Zero outliers in the blood batch and four outliers in the muscle batch were identified and expression profiling was repeated for these patients. For the analysis we used only the repeated profiles.

The study was approved by the ethics committee of the Faculty of Medicine at LMU Munich, Germany; University of Rochester Research Subjects Review Board, Rochester, New York and University of Florida Institutional Review Board, Gainesville, Florida, USA.

It should be noted that our dataset differs from the dataset collected by Nakamori *et al.* in the following ways:

1. We did not perform confirmatory RT-PCR analyses.

2. Our dataset includes both blood and muscle tissue samples.

3. The number of mRNA-profiled participants is about twice the size of the discovery dataset of Nakamori *et al.*, however, we did not perform a follow-up validation study.

4. Unlike Nakamori *et al.* we have additional information to participant's case-control status, specifically, the mode of the CTG repeat length from blood (MAL). We also estimate the repeat length at birth (Progenitor Allele Length, PAL) using a previously developed method [18].

5. We sample from the same muscle group (quadriceps), as opposed to from a wide range of muscle groups (biceps, quadriceps, tibialis anterior, diaphragm, vastus lateralis) for all study participants, which eliminates a potentially important confounding effects.

6. All our participants *were* alive at the time of sample collection, which eliminates another potential confounder, but other potential confounders still remain, see Limitations.

The dataset is deposited to Array Express with accession number E-MTAB-7983.

## Affymetrix GeneChip™ human exon 1.0 ST microarray

The microarray chip used was the Affymetrix GeneChip™ Human Exon 1.0 ST microarray, which contains over 5 million probes, *i.e.* short cDNA sequences, which target transcribed regions with high specificity. Each probe in the HuEx chip contains precisely 25 nucleotides. Consequently, transcribed regions of interest, which are shorter than 25 nucleotides, for example some of the short exons, are not targeted. This is an important limitation in the context of DM1, as some AS events previously described in DM1, such as AS of exon 5 in cardiac troponin T (*TNNT2*) [22] cannot be detected.

Each continuous section of DNA can be targeted by a collection of up to four probes, referred to as probeset. DNA sections targeted by the chip include known and suspected coding exons in known and suspected genes, as well as non-coding genomic features, including 5' and 3' UTRs, and various other types of transcribed or hypothetically transcribed DNA (miRNA, rRNA, pseudo-genes, *etc.*). All probes targeting such a region belong to a single probeset. Probesets are further grouped into transcription clusters, which correspond to the entire genes.

## Data preparation and analysis

We designed and built a pipeline, programmed in Python, which has the following data preparation capabilities: Reading raw Affymetrix CEL v4 files (peer reviewed and merged into Biopython [23, 24]); quantile normalisation and log2 transformation of intensity data; strict annotation of Affymetrix probes using GENECODE v26 lift 37 through selecting probes corresponding to annotated GENECODE transcripts of type "protein_coding", annotated genes of

type "protein_coding" and exons of type "CDS" or "UTR". Appendix S1 Appendix. gives full source code, user manual and additional explanation of each step of this pipeline.

The pipeline's final output are two directories: "experiment_muscle" and "experiment_-blood", each containing 19,826 files, with each file corresponding to a single gene (e.g. TNNI1) whose filenames correspond to HGNC gene names. The following is a two-line excerpt from one of such files, "experiment_blood/TNNI1". Data for several patients has been removed to enhance clarity:

```
gene_name        probeset_id        seq5to3plus        chrom        strand
↪        genecode_left        genecode_right        x        y
↪        patient_111747589        patient 117440822                . . .
↪        patient_896445336
TNNI1    2450836 TGGCCTGTGCCTCGCCGTAGACTGC chr1—
↪                201390827                201390851    195    793
↪        6.69115092487    6.44442470429        . . .        6.86989190909
```

In each file, the first row is a header containing tab-separated names of data or metadata types contained in a given column. Below we give a brief description of some of the data or metadata types:

- seq5to3—unlike Affymetrix we always report the sequence in 5' to 3' direction, and always with regards to the plus strand, even if the coding sequence is contained on the minus strand.

- genecode_left, genecode_right—these are genomic coordinates as reported by a reference assembly (GRCh37). Following convention, first coordinate is 1-based, and coordinates are left-, right- inclusive.

- x, y give the x and the y coordinates of probes on the chip.

- patient_*—these are quantile normalised and log2-transformed intensities at the given probe for the given study participant.

  Each subsequent row contains data and metadata for a single probe.
  We develop a predictive model, which closely follows that of Lee *et al.* [17].

At a conceptual level, the model takes probe intensities (records under the patient_* headers) as the model's input. It then predicts the CTG length as the model's output. Throughout the manuscript we often refer to the model's predictions as "predicted MAL" or "effective repeat length". The effective repeat length is a single numeric value, which allows us to capture the extent of alternative splicing defects using a familiar concept, i.e. the length of the disease-causing CTG expansion.

The model works with pre-selected sets of genes that act as candidate biomarkers. For this purpose, we look at the following collections of genes:

1. A previously identified selection of genes, listed in S2 Appendix and identified by Nakamori et al. [8] as genes whose AS is disrupted in DM1. We codename these genes "DM1-AS".

2. A previously identified selection of genes, listed in S3 Appendix and identified by Batra et al. [9] as genes whose AP is disrupted in DM1. We codename these genes "DM1-APA". The overlap of this list of genes with DM1-AS is a list of two genes: LDB3, MBNL2.

3. A single gene (which also belongs to DM1-APA), Troponin I1, slow skeletal type, TNNI1. We codename this single-gene collection "TNNI1". The gene was chosen *post-hoc*.

4. All human genes, as identified in data preparation step. We codename this collection "ALL".

Then we execute eight separate statistical analyses (four muscle and four blood analyses), based on these groups of genes. Each analysis was carried out as follows:

1. We randomly split our data into two sets, the training and the testing set, each containing 70% and 30% of participants respectively. We restricted our analysis only to genes, which belonged to a particular group of genes being studied (DM1-APA, DM1-AS, TNNI1, ALL)

2. Following Lee *et al.*, we select (up to) 500 probes whose intensities across all patients in the training set are most correlated with their corresponding MAL. We work at the level of individual probes, which allows us to circumvent issues around GC-correction and probe-set aggregation. Probe data are fed directly into the model.

3. Again using the training set, we trained a 2-dimensional Partial Least Squares Regression (PLSR) model on selected probes as features and corresponding MAL as the model output, which we later used to predict MAL in the testing set, again following Lee *et al.*.

4. We repeated steps 1 to 3 10,000 times.

5. We report coefficient of determination ($R^2$) of the predicted MAL with the measured MAL across all folds obtained in step 4. Given that both the predicted MAL and the measured MAL are single numeric values, the computation of $R^2$ does not suffer from issues relating to a high number of predictors.

6. We simulated a distribution of $R^2$ of a random predictor, and obtained a p-value for the prediction of $R^2$ in step 5.

To confirm that any observed signal is not a by-product or an artifact of the mathematical model used, or its implementation, we carried out the same kind of analysis using three other mathematical models:

1. lasso

2. random forest regression

3. linear regression

To relate our findings to a potential clinical setting, we present power analysis relating various potential treatment effect sizes to their detectability in future clinical trials. The power analysis was performed as follows:

1. We simulated the distribution of MAL prediction error for the best predictor. This was achieved in a similar way to the computation of the coefficient of determination: we used the same method of splitting our data into a training and a testing set, selecting features, training the model and making predictions. The only difference was that instead of computing $R^2$, we used the predicted MAL set to compute the prediction error.

2. We simulated the distribution of predicted MAL post-treatment, with the assumption that predicted MAL would be reduced by 10%, 20% or 50%, which correspond to, respectively, small, medium and large treatment effect size.

3. We established the rejection region for a null hypothesis of "no treatment effect" at $\alpha = 0.05$.

4. We simulated the power, $(1 - \beta)$, for studies involving 10 to 200 participants, in increments of 10 participants.

## Results & discussion

### Performance of PLSR-based MAL predictors in muscle and blood

Using our predictive model based on PLSR and a selection of candidate biomarkers: DM1-AS; DM1-APA; TNNI1 and ALL, we can report the following capabilities to predict MAL from mRNA profiles.

In muscle we can detect strong signal for some of the selected candidate biomarker sets, with the two strongest predictors, DM1-AS and TNNI1, giving us $R^2$ and p-values equal to 0.285, 0.322 and 0.0046, 0.0023 respectively. All values of $R^2$ and p-values, as well as Root Mean Square Deviation (RMSD) are given in Table 1. In particular, DM1-AS looks promising as a set of biomarkers, giving weight to the findings of Nakamori *et al.* [8].

Based purely on numerical analysis of obtained results, one could declare TNNI1 as the best predictor of MAL in muscle, but care must be taken, as we chose this particular gene post-hoc and its true predictive value might have been influenced by issues related to multiple hypotheses testing. An additional validation study would have to be performed before drawing any conclusions on the performance of TNNI1 as a DM1 biomarker.

A safer belief can be assigned to the predictive value of "DM1-AS" and "DM1-APA" as both sets of genes have been previously implicated in DM1 [9], and we can detect strong signal from both sets of genes, with models based on these genes capturing respectively 28.5% and 15.1% of MAL variance in our study group.

One has to note an interesting observation relating to the "curse of dimensionality" and the performance of our PLSR-based model. Although TNNI1 is a subset of DM1-APA, TNNI1 on its own is a much better predictor than DM1-APA, as increasing the number of genes (features) increases the dimensionality of our data and worsens the prediction delivered by the PLSR model.

A single run of 10,000 repetitions of a simulation can be visualised by plotting the predicted value of MAL against the actual, adding a small amount of random noise on the x-axis. For DM1-AS in muscle, such visualisation is given in Fig 1.

In blood we cannot detect significant signal for most candidate biomarkers, with $R^2$ not significantly respectively greater and lower than would be expected by chance, except in one case, DM1-APA, whose $R^2$ and p-value are 0.15 and 0.0564 respectively. Full $R^2$ and (uncorrected) p-values are given in Table 2. Poor performance of blood data acting as a predictor of repeat length is perhaps not unexpected, as all candidate biomarkers evaluated in this study are based

**Table 1. 10000 repetitions of a simulation predicting MAL from muscle cross-validated with a testing set separate from the training set.**

|         | DM1-AS  | DM1-APA | TNNI1   | ALL     |
|---------|---------|---------|---------|---------|
| $R^2$   | 0.285   | 0.151   | 0.322   | 0.085   |
| p-value | 0.0046  | 0.0455  | 0.0023  | 0.1396  |
| RMSD    | 327.821 | 361.443 | 326.266 | 370.429 |

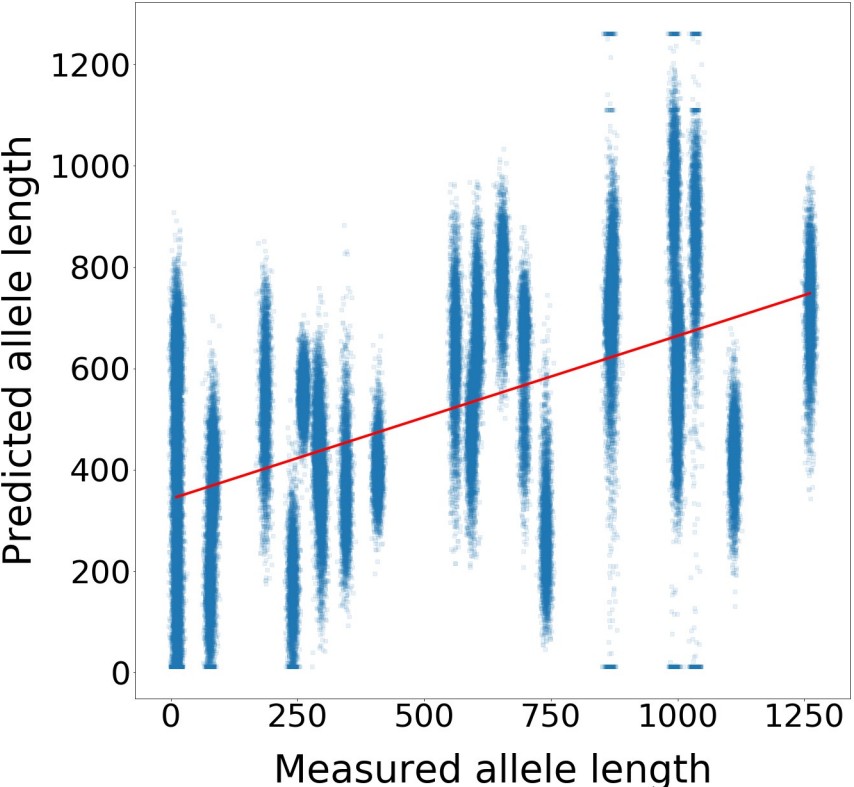

**Fig 1. DM1-AS muscle MAL prediction.** 10,000 repetitions of cross-validated MAL prediction from genes labeled DM1-AS from muscle for 18 training samples.

on prior analyses of muscle data, and might not be expressed, or only lowly expressed in blood, or that expression profiles might be very different from expression profiles in muscle. Only DM1-APA can potentially carry some predictive value, however, not as strongly as in muscle.

## Other models

As described in Data preparation and analysis, we have used a model based on feature selection from sets of candidate mRNA biomarkers to predict the MAL of DM1 CTG repeat using PLSR. As reported in Performance of PLSR-based MAL predictors in muscle and blood, the set of genes DM1-AS is the strongest predictor if we limit our consideration to predictors chosen *a-priori* (*i.e.* excluding TNNI1).

A potential source of criticism could be that the effect observed is a technical effect due to the choice or the implementation of the mathematical model used (PLSR). We thus re-ran the analysis as described before in Data preparation and analysis using three mathematically distinct models: lasso, random forest regression and linear regression.

**Table 2. 10000 repetitions of a simulation predicting MAL from blood cross-validated with a testing set separate from the training set.**

|  | DM1-AS | DM1-APA | TNNI1 | ALL |
|---|---|---|---|---|
| $R^2$ | 0.044 | 0.104 | 0.049 | 0.011 |
| p-value | 0.2239 | 0.0586 | 0.2014 | 0.5506 |
| RMSD | 363.334 | 344.767 | 392.414 | 371.649 |

**Table 3. 10000 repetitions of a simulation predicting MAL from muscle, using DM1-AS as a predicting set and a selection of mathematical models.**

|         | linear regression | PLSR    | lasso   | random forest |
|---------|-------------------|---------|---------|---------------|
| $R^2$   | 0.291             | 0.285   | 0.286   | 0.149         |
| p-value | 0.00418           | 0.00457 | 0.00454 | 0.0478        |

We report the performance of these models in Table 3:

It should be noted that all models pick up statistically significant signal with both PLSR, lasso and linear regression performing almost equally well, and random forest performing about two times poorer (but still significantly better than a random predictor), which allows us to conclude that the effect observed is unlikely to be a modeling artifact.

## Qualitative characterisation of the strength of observed correlations

The strength of observed correlations of CTG repeat length and effective CTG length as measured by AS defects needs to be put in the context of similar correlations observed in other studies. We should distinguish here studies looking at large cohorts with a wide variety of clinical symptoms, e.g. studies including both congenital and late-onset participants, from studies looking at a relatively narrow range of participants, e.g. adult-onset, ambulatory participants only. In the former, highly variable cohorts, CTG repeat length has been shown to explain a large fraction of the variance of such DM1 symptoms as age-at-onset (e.g., $R^2$=0.640 in Morales et al. [18]). It has been much less successful in the latter type of studies, where study selection criteria indirectly limit the range of observed CTG repeats.

The study described here is of the latter type, with selection criteria being adult-onset, ambulatory DM1 patients. In this context, and taking into account the modest size of this study, the most significant correlation, i.e. between effective CTG length as predicted from DM1-APA and the true CTG length as measured using PCR ($R^2$=0.29) actually compares favourably with symptoms most strongly correlated in similar studies, such as correlation of CTG repeat length and grip strength ($R^2$=0.443), pinch strength ($R^2$=0.419) and ankle dorsiflexor strength ($R^2$=0.202) [19].

A tempting question to ask is whether the study presented here allows us to draw conclusions with regards to explainability of AS defects with CTG length. Unfortunately, the study design did not include any biological or technical replicates. As a result, we were unable to quantify the relative contributions of different factors to overall variance in predicted CTG length. An extended discussion of this theme is present in section Limitations.

## Applying the model in a clinical setting

Let us now consider the potential application of this predictive model in the context of evaluating efficacy of a DM1 treatment. DM1 patients would undergo muscle biopsy before starting the treatment, and another biopsy after the treatment had been started and necessary biological changes to reverse DM1 symptoms had occurred. Both biopsies would be mRNA profiled, and the resulting profiles would be used to perform MAL predictions. We expect that pre-treatment prediction would correspond to the actual MAL of any given participant. We expect that post-treatment prediction of MAL would correspond to an "effective MAL", which we would expect to be lower than the "actual MAL" in affected participants, as long as the treatment is effective and DM1-induced disruption of AS or APA, as measured by DM1-AS or DM1-APA biomarkers is measurably reversed in obtained mRNA profiles. Pre-treatment and post-treatment predictions could be combined into a statistic that, given enough patients,

would allow us to quantify the efficacy of the treatment at the molecular level. We discuss this idea further in the chapter Power Analysis.

In some respects, such a study could allow for better performance of the model, conceptually, it should be easier to capture DM1 specific expression changes in a setting where noise due to varied genetic backgrounds of participants can be reduced by looking at pairs of measurements of a single participant. There are a number of details in such study design, which need to be discussed by the community and decided upon, among others:

1. The mathematical basis of the model used. We propose a PLSR-based model, and demonstrate that models based on lasso and linear regression perform similarly, but other models can also be considered, in particular the work of Azencott *et al.* [16] in the context of $L_1$-penalised regression (lasso) looks promising as it allows to incorporate prior biological knowledge, in the form of protein-protein interaction networks or other types of graph ontology, into the model, through the introduction of additional penalties based on discrete Laplacian, or apply alternative modelling strategies based on network flow, thereby increasing its predictive power.

2. The training/testing pre-treatment/post-treatment data split and which genes should be included in the model input.

3. The size of the study. Predictions in our study are based on 24 participants for blood and 18 for muscle. How much would the models' predictive power improve with a larger dataset?

4. Establishing the clinically relevant effect size. Pandey *et al.* [7] report various efficacies of a candidate DM1 treatment ISIS 486718 to lower toxic *DMPK* concentrations in wild-type and transgenic animal models and a range of tissues, starting with the efficacy of about 50% in cardiac muscle, through about 70% in skeletal muscle, up to about 90% in liver and skeletal muscle. However, measuring *DMPK* levels may not necessarily directly correspond to the efficacy of treatment to reverse symptoms, as the relationship between the quantity of the toxic transcript, splicing disruption and eventual clinical symptoms may be complex and non-linear. Conservatively, we need to expect the rate of symptom reduction to be lower than the reported 50% to 90%. A difficult open question is what minimum treatment efficacy we are willing to accept as clinically significant?

We would also like to note that the statistical framework proposed here is not limited to the experimental technique used. The framework could equally well be applied to new data sets, including RNA-seq, as long as they feature a sized CTG repeat for each participant in the dataset.

## An informal setting to explore the dataset

As this work is the first presentation of the DMBDI dataset, we recognise that further work might build on the dataset in ways which differ from our approach and cannot be predicted at the current stage of our understanding of DM1-related AS/APA changes. To facilitate this, we would like to propose a tool which on one hand allows for informal, interactive and exploratory analysis of the dataset and on the other allows the flexibility of building a custom analysis —just like the one presented here.

The tool is available online [25], and is implemented as a jupyter notebook with custom visualisation of filtered and normalised DMBDI data. The flexibility of the tool comes at a cost. In order to support arbitrary bioinformatics analyses, we have to support arbitrary code execution, which in turn requires protecting the tool with a password. We will share the password

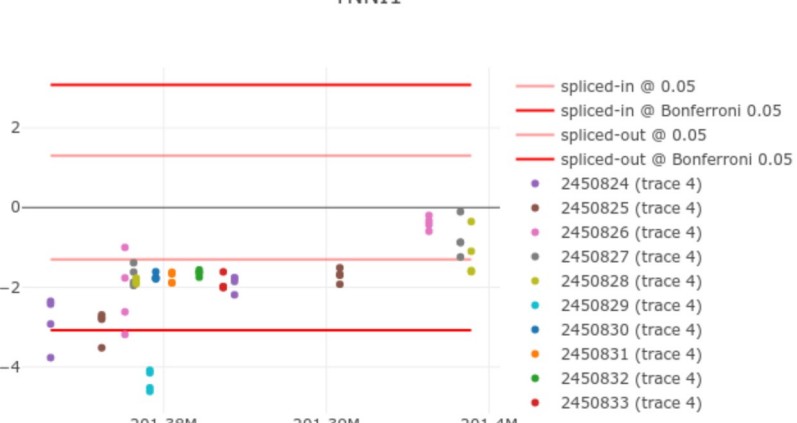

**Fig 2. TNNI1 railway plot shows an APA event at probeset 245089.**

with any *bona fide* researcher upon request. A walk-through video showcasing the capabilities of the tool is available on youtube [26].

The major capability of the tool is the ability to produce "railway plots". Railway plots introduce the idea of a Manhattan plot from genomics community into transcriptomics. Each point represents statistical significance of the change of expression signal at a single probe across DM1 spectrum, as supported by experimental data. Points which belong to the same probeset, are identically coloured. See Fig 2 for an example railway plot.

In a railway plot the y-axis represents negative logs of p-values of a two-tailed test against a null hypothesis of no expression change at a single probe across the DM1 spectrum. Fig 3 visualises one of such linear regressions for a probe belonging to probeset 245089. The logs of p-

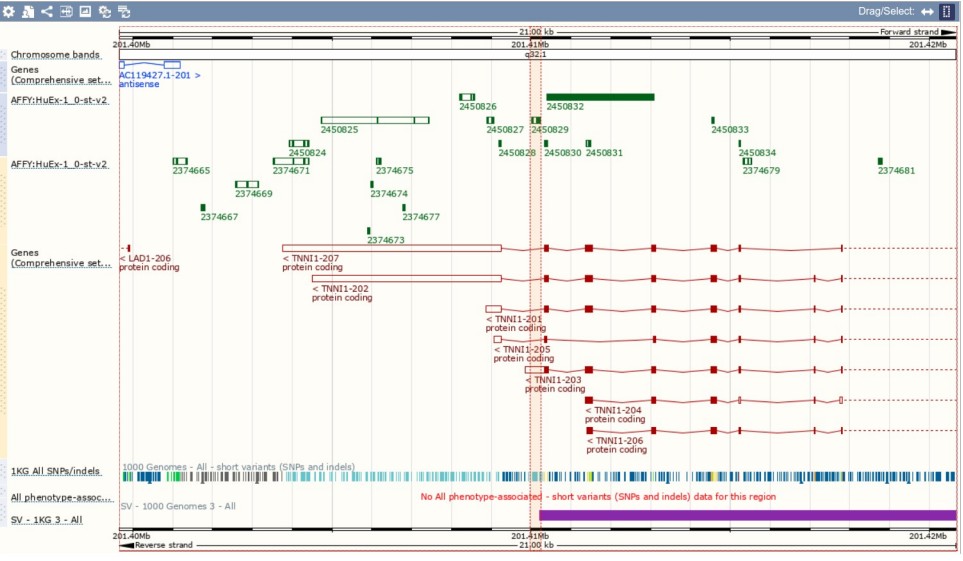

**Fig 3. Linear regression of expression intensity at a single probe belonging to probeset 245089 against DM1 repeat length.**

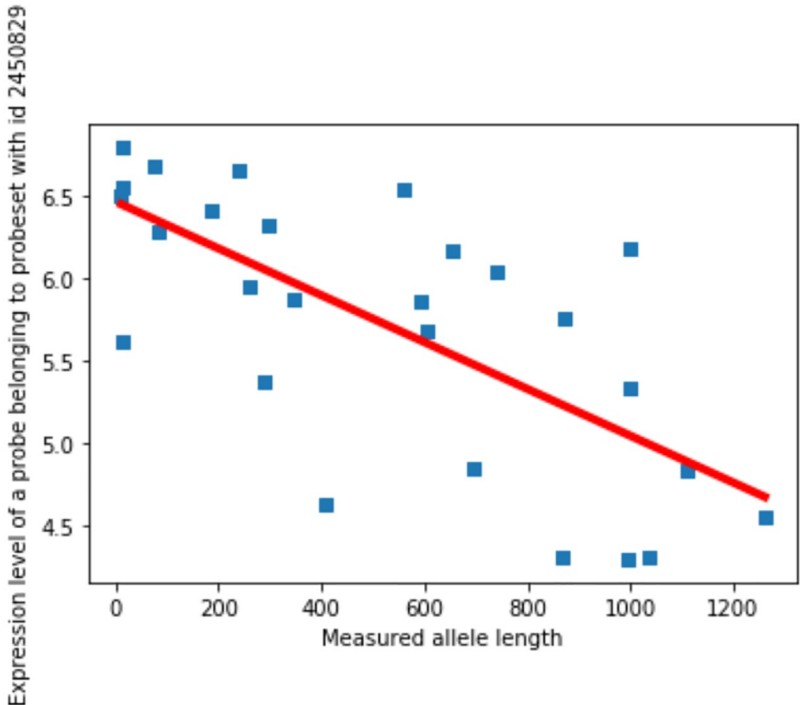

**Fig 4. Visualisation of genomic coordinates of TNNI1 transcripts using Ensembl.**

values are signed in accordance with the direction of the slope of the regression, with negative values indicating splice-out and positive values indicating a splice-in type of event.

For each splice-out and splice-in directions in the plot, we show thresholds of statistical significance. The first pair of thresholds (faded red and blue respectively) correspond to statistical significance threshold of 0.05. The second pair of thresholds correspond to $0.05/n$, where n is the total number of probes in the plot (saturated red and blue respectively). This is analogous to the way Manhattan plots are often presented for GWAS.

Turning our attention from statistics to biology, we can ask about a likely biological interpretation of an observed splice-out event detected at probeset 2450829. Using Ensembl [27], see Fig 4), we can identify transcript TNNI1-203, which is the only GENECODE transcript featuring a probe selection region targeted by the probeset 2450829, in form of an alternative 3' UTR. This allows us to suggest that TNNI1-203 is downregulated in participants with longer DM1 repeats. GENECODE annotation further informs us that TNNI1-203 is an APPRIS P2 transcript (i.e. a candidate principal variant, a designation which comes from high level of support for functionality of the isoform), and is not a CDS incomplete transcript, which allows us to strengthen our belief in the fact that this is a biologically functional/ protein-coding transcript, which can play a role in the DM1-related AS/APA changes.

Finally, and returning back to statistics, we can ask whether high significance of the splice-out event is a result of multiplicity effect, given that the gene was chosen *post-hoc* from a pool of candidate biomarker genes as determined by Batra et al. and Nakamori et al. [8, 9]. A standard approach here would be to combine the data from the discovery dataset with the data from the replication dataset, compute a more powerful test, and apply multiplicity correction. This is not possible in this case as the discovery dataset, underlying both studies is a case-control dataset, whereas our dataset captures DM1 status as a continuous variable via DM1 CTG

**Table 4. Power analysis.** Entries in the table report power to detect treatment effect based on the size of a cohort (from 10–200 participants) and the treatment effect of the study to reverse splicing changes (10, 20 and 50%). Entries denoting power greater than 95% are presented in boldface.

| study size (participants) | treatment effect 10% | treatment effect 20% | treatment effect 50% |
|---|---|---|---|
| 10 | 0.100 | 0.209 | 0.635 |
| 20 | 0.131 | 0.322 | 0.855 |
| 30 | 0.162 | 0.423 | 0.947 |
| 40 | 0.193 | 0.517 | **0.981** |
| 50 | 0.226 | 0.596 | **0.994** |
| 60 | 0.259 | 0.671 | **0.998** |
| 70 | 0.283 | 0.723 | **0.999** |
| 80 | 0.311 | 0.772 | **0.9998** |
| 90 | 0.339 | 0.816 | **0.99998** |
| 100 | 0.370 | 0.851 | **0.99998** |
| 110 | 0.396 | 0.881 | **0.99999** |
| 120 | 0.418 | 0.901 | **1.0** |
| 130 | 0.448 | 0.924 | **1.0** |
| 140 | 0.467 | 0.937 | **1.0** |
| 150 | 0.500 | **0.952** | **1.0** |
| 160 | 0.523 | **0.962** | **1.0** |
| 170 | 0.545 | **0.969** | **1.0** |
| 180 | 0.568 | **0.977** | **1.0** |
| 190 | 0.589 | **0.981** | **1.0** |
| 200 | 0.613 | **0.986** | **1.0** |

repeat length measurement. An alternative is to combine both p-values using, e.g. Fisher's method [28], which shows that the combined p-value against the null hypothesis of no change of signal intensity at the probeset 2450829 is $4.19 \times 10^{-10}$. Computing the Bonferroni correction with multiplicity factor of 1.4 million, equal to the total number of HuEx probesets [29], shows that the combined p-value based on the discovery dataset and our dataset is $5.88 \times 10^{-4}$. This is a strong confirmation of the significant correlation of DM1 status with TNNI1-203 downregulation. Details of the computation are available in S1 Appendix.

## Power analysis

Our final contribution, which is of critical significance in the context of any future clinical trial is a power analysis of the current model. We report power for a selection of possible treatment effect sizes (10%, 20% and 50% reduction in effective MAL on a per-patient basis) and a selection of study participants. Our power is defined as $(1 - \beta)$, where $\beta$ is a supremum of the probability of committing a type II error, with the supremum of the probability of committing type I error ($\alpha$) kept at a constant 0.05. Table 4 reports power, $(1 - \beta)$, to detect treatment effect of a two-tailed test with p-value cut-off of 0.05 (0.025 per tail), with the statistic simulated from MAL predictions of our model, for varying treatment effect and study sizes. We try to keep our cohort sizes realistic for a rare disease, *i.e.* we allow for patient numbers to range from 10 to 200.

Ideally we would like to be able to achieve power of more than 95%, even with small treatment effect sizes and a small number of patients, but our model, trained on 18 participants, doesn't allow for such level of control over type II error for all but medium or large treatment effects (more than 20% and 50% respectively) and large (more than 140 participants) or medium-sized (more than 30 participants) clinical trials respectively.

However, our results combined with expected improvements of the model performance due to larger training samples, and better gene selections, such levels might be reached for medium treatment effect size (20% reduction in effective MAL) and large clinical sizes.

## Limitations

**Tissue sampling.** A source of potential criticism is that muscles of DM1 patients have physiological differences (atrophy, increased fat content), especially when disease is severe. Quite possibly observed changes in AS/APA are partly attributable to these physiological differences in DM1 as opposed to purely biomolecular differences. The structure of this counter-argument could be as follows:

Muscles of DM1 patients have higher fat content than affected controls. Muscle samples collected from DM1 patients have higher ratio of intermuscular adipocytes to myocytes. Adipocytes have different AS/APA profiles than myocytes. Observed AS/APA changes in the DM1 spectrum are mostly derived from differences in adipocyte/myocyte profile. As a result, mRNA study of muscle tissue is no more effective (and possibly less effective) than a blinded study based on pathophysiological inspection of the tissue.

This argument can, of course, be extended to other physiological changes than increased fat content, and other molecular events than AS/APA. Bachinski et al. [30], for example, propose that splicing changes could be a secondary result of muscle regeneration.

There are multiple ways to address these concerns:

1. Alternative predictive model design based on tissue culture models, where *in vivo* limitations are reduced, with homogeneity of the cellular composition of the model being a big advantage.

2. Investigating methods to correct for potential "physiological" covariates (*e.g.* fat content), using purely statistical techniques to estimate covariate influence from gathered data and existing prior information (*e.g.* mRNA profiles of adipocytes), or biological methods, such as a recent effort to collect higher quality muscle samples through MRI-guided biopsy [31].

3. Discovering DM1 biomarkers in blood, as opposed to muscle. Blood, being much more homogeneous tissue than muscle is expected to be less prone to the existence of confounding variables. Additionally, necessity of muscle sampling was highlighted to be a "main drawback" [8]. However, achieving this would require at least two separate, successful studies, one to identify biomarkers and one to evaluate them. Even if blood biomarkers were identified, their clinical utility might be limited, as a reduction of an effective MAL in blood would not be as direct evidence of treatment effectiveness as such reduction in muscle.

4. Introduction of positive controls in experimental designs. Biological samples in the positive control group would be composed of tissue collected from individuals with other muscular dystrophies (e.g. Becker muscular dystrophy, Duchenne muscular dystrophy, facioscapulo-humeral muscular dystrophy or tibial muscular dystrophy). These diseases feature dystrophy and increased muscle regeneration program as part of their phenotype, but without disruption of RNA-binding splicing factors. Absence of DM1-specific splicing changes in these positive controls would allow to rule out alternative explanations of mis-splicing mechanisms (e.g. muscle regeneration) and strengthen our belief in currently accepted models of molecular pathomechanism of DM1.

5. Confining the analysis to transcripts which are exclusively or predominantly expressed in skeletal muscle, not fat.

**Weak correlations of predicted and true CTG length.** Correlations described in the study are quite modest, with effective CTG length as predicted from DM1-APA and the true CTG length as measured using PCR giving us the highest correlation with r2 = 0.29. We would like to group potential reasons for this into two categories of factors, which capture the most likely reasons for weak correlations: 1. Weak correlation of CTG repeat length with progressive DM1 symptoms in general (potentially including the extent of AS defects). This highlights a shortcoming of our current understanding of molecular pathology of DM1. 2. Study design, and in particular experimental techniques used in the study introducing large amounts of technical noise. While the first factor is well known [3], and a deeper understanding of DM1 pathomechanism, e.g. through further study of the RNA binding proteins (RBPs) is warranted, the study design (including experimental factors) is likely to be a major contributing factor. Unfortunately, the study design did not include any biological or technical replicates, so we were unable to quantify the relative contributions of different factors to the error terms present in both measurements, and especially the measurements of AS defects. Consequently, we are limited to a primarily qualitative discussion of the study limitations:

1. We are studying a complex, noisy system: atrophied muscle of individuals who volunteered in the study. These individuals exhibited difference in age (which would contribute to somatic mosaicism discrepancy between blood/muscle), sex, and would have different lifestyles. All these factors can contribute to DM1 pathomechanism in a way, which couldn't be captured with CTG repeat sizing in blood.

2. The platform we are using (Human Exon array) cannot detect all available signal (e.g. some of the short exons are not targeted). There are several other issues with the experimental setup: measurements are based on just 9 pixels from the original image from the scanner. Most probes have no technical replicates. The array used was not a junction array, it's difficult to attribute within-transcript change to any particular AS events/defects. Probe GC content can be a confounder (but the method we've used is resistant to any additive bias from GC content). Dyes used in microarrays are very sensitive to daily conditions in the laboratory, e.g. it has been shown that levels of atmospheric ozone has significant effect on the measurements [32]

3. Small sample size. In the study we obtained a complete tissue collection from only 27 individuals. Intuitively, this is a small sample size when judged by a typical size of a phase 1 clinical trial. Simulation confirms the intuition that a better model could be obtained by simply training on a larger dataset. Specifically, R2 of 28.5% was reported for a model which used 18/9 as the training/testing split for the 27 individuals. Using the same model with 14/13 as a splitting strategy, performance of the model measured R2 to be only 17.6%. We would like to note that our estimate is conservative, and a larger sample size would likely give us a higher predictive strength.

4. The original list of biomarkers was discovered in different muscle groups.

5. Our model is quite simple—it doesn't encode biological intuition behind alternative splicing or alternative polyadenylation. All features are treated as independent, even within a single transcript.

6. The repeat was sized in blood as opposed to muscle. Repeat sizing in muscle is very challenging, but would likely give stronger levels of signal.

## Conclusion

In this study we design and build a model based on PLSR, which can explain as much as 28.5% of the variance in DM1 CTG trinucleotide expansion from mRNA splicing data. Such explainability is only obtained when the model is trained on expression data from genes previously identified by Nakamori *et al.* [8] as having disrupted AS on data obtained from muscle samples. We show how such model could be used in a clinical setting in the context of emerging DM1 treatments, and report power analysis to detect treatment effect depending on size of the treatment effect, type 1 error ($\alpha$) and potential size of the clinical trial.

## Supporting information

**S1 Appendix.**
(PDF)

**S2 Appendix.**
(PDF)

**S3 Appendix.**
(PDF)

## Acknowledgments

We would like to thank the Marigold Foundation for organising DMBDI.

We would like to thank Linda Bachinsky and Keith Baggerly for their helpful comments on multiple revisions of this manuscript, as well as performing initial analysis and quality control on blood and muscle microarray samples. We would also like to thank Anna Casasent (nee Unruh) for performing quality control on blood and muscle microarray samples.

## Author Contributions

**Conceptualization:** Adam Kurkiewicz, Ralf Krahe, Jack Puymirat, Benedikt Schoser, Lubov Timchenko, Tetsuo Ashizawa, Charles A. Thornton, Simon Rogers, Darren G. Monckton.

**Data curation:** Adam Kurkiewicz, Ralf Krahe, Darren G. Monckton.

**Formal analysis:** Adam Kurkiewicz.

**Investigation:** Anneli Cooper, Emily McIlwaine, Sarah A. Cumming, Berit Adam.

**Methodology:** Adam Kurkiewicz.

**Project administration:** Ralf Krahe, Simon Rogers, John D. McClure, Darren G. Monckton.

**Resources:** Jack Puymirat, Benedikt Schoser, Tetsuo Ashizawa, Charles A. Thornton.

**Software:** Adam Kurkiewicz.

**Supervision:** Simon Rogers, John D. McClure, Darren G. Monckton.

**Validation:** John D. McClure.

**Visualization:** Adam Kurkiewicz.

**Writing – Review & Editing:** Charles A. Thornton, Darren G. Monckton.

**Writing – original draft:** Adam Kurkiewicz.

**Writing – review & editing:** Ralf Krahe, Simon Rogers.

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
