## [Decision Letter · Decision Letter 0]

22 Oct 2019

PONE-D-19-24916

Towards development of a statistical framework to evaluate myotonic dystrophy type 1 mRNA biomarkers in the context of a clinical trial

PLOS ONE

Dear Mr. Kurkiewicz,

Thank you for submitting your manuscript to PLOS ONE. After careful consideration, we feel that it has merit but does not fully meet PLOS ONE’s publication criteria as it currently stands. Therefore, we invite you to submit a revised version of the manuscript that addresses the points raised during the review process.

We would appreciate receiving your revised manuscript by Dec 06 2019 11:59PM. To enhance the reproducibility of your results, we recommend that if applicable you deposit your laboratory protocols in protocols.io, where a protocol can be assigned its own identifier (DOI) such that it can be cited independently in the future. For instructions see: http://journals.plos.org/plosone/s/submission-guidelines#loc-laboratory-protocols

We look forward to receiving your revised manuscript.

Kind regards,

Ruben Artero, Ph.D.

Academic Editor

PLOS ONE

**Journal Requirements:**

"Collection, data analysis and public data release of human samples were approved by Institutional Review Boards of clinicians who contributed to this study, as arranged with the Marigold Foundation.  The contributing clinicians were: Jack Puymirat, Benedikt Schoser, Tetsuo Ashizawa and Charles Thornton.  Anonymised data from the study has been in the public domain since 31st of May 2019: " ext-link-type="uri" xlink:type="simple">https://www.ebi.ac.uk/arrayexpress/experiments/E-MTAB-7983/"

"Adam Kurkiewicz

Declares ownership of Illumina and PacBio shares.

Anneli Cooper

Has served on a Scientific Advisory Board for AstraZeneca (Trypanosomiasis).

Sarah Cumming

None declared

Berit Adam

None declared

Ralf Krahe

None declared.

Jack Puymirat

None declared

Benedikt Schoser

Benedikt Schoser is member of the Neuromuscular advisory board of Audentes Therapeutics, USA, and Scientific advisory of Nexien BioPharm, USA. He received speaker honoraria from Sanofi Genzyme, Amicus Therapeutics, Lupin Pharmaceuticals, and Kedrion. He received an unrestricted research grant from Sanofi Genzyme USA (2016-2019), Greenovation FRG (2017-2020), and from the Marigold foundation (2014)

Lubov Timchenko

None declared.

Tetsuo Ashizawa

1. My wife has stocks and stock options of BIOPATH Holdings, Inc.

2. One US and international patent approved.

3. Grants from NIH, the Myotonic Dystrophy Foundation, the National Ataxia Foundation and Biogen.

4. I am a member of the advisory board for the National Ataxia Foundation and that for the Myotonic Dystrophy Foundation.

Simon Rogers

None declared

John McClure

None declared

Darren G Monckton

Professor Monckton has been a scientific consultant and/or received an honoraria or stock options from Biogen Idec, AMO Pharma, Charles River, Vertex Pharmaceuticals, Triplet Therapeutics, LoQus23, BridgeBio, Small Molecule RNA and Lion Therapeutics. Professor Monckton also had a research contract with AMO Pharma and CHDI has received research grants from the European Union, European Huntington Disease Network, National Institute of Health, Muscular Dystrophy UK and the Myotonic Dystrophy Support Group.

Professor Monckton is on the Scientific Advisory Board of the Myotonic Dystrophy Foundation, is a scientific advisor to the Myotonic Dystrophy Support Group and is a vice president of Muscular Dystrophy UK.

Charles Thornton

Dr. Thornton has received sponsored research support from Ionis Pharmaceuticals, Biogen, Genzyme, and Dyne Therapeutics, and research grants from the  National Institutes of Health, Food and Drug Administration, Muscular Dystrophy Association, and Myotonic Dystrophy Foundation.  Dr Thornton has served on the Scientific Advisory Board for Dyne Therapeutics.  "

**Additional Editor Comments (if provided):**

Despite of interest to the journal, reviewers have found a number of significant shortcomings in the manuscript that need to be addressed by the authors before a new assessment by the experts.

**Comments to the Author**

1. Is the manuscript technically sound, and do the data support the conclusions?

Reviewer #1: Yes

Reviewer #2: Partly

2. Has the statistical analysis been performed appropriately and rigorously? 

Reviewer #1: Yes

Reviewer #2: No

3. Have the authors made all data underlying the findings in their manuscript fully available?

Reviewer #1: Yes

Reviewer #2: Yes

4. Is the manuscript presented in an intelligible fashion and written in standard English?

Reviewer #1: Yes

Reviewer #2: Yes

5. Review Comments to the Author

Reviewer #1: I have several specific suggestions for the authors

1. The discussion of “effective repeat length” on page 5 could use better framing. The authors are referring to the fact that there is a relationship between repeat length and degree of splicing changes, and that in fact the most important therapeutic read out is return of correct AS vs reduction of the CTG repeat length. This should be more explicitly stated.

2. More recent work has used RNA-seq to identify splicing changes (such as Wagner et al.). Can the authors provide a rationale for reverting to the microarray approach?

3. It is not clear from the informatic approach what steps the authors took to correct for multiple comparisons? If not, this should be provided as a limitation.

4. In the results, the authors do not comment on the strength of the correlation. Even though they are statistically significant, correlations 0.4 are modest at best. Please provide a qualitative rating of the correlations.

5. It is as interesting, if not more, that only 29% of the AS defects seen can be explained by the repeat length alone. This is a strong finding given the statistical significance of the R2 in the results. Yet the authors do not comment on the biological relevance. It suggests that other components of the RBP complex may provide equal contributions to the weight of the pathology. Another alternative is that a minor fluctuation in MAL via a therapy would not be sufficient but rather significant knockdown would be required to repair AS. In any case, there should be discussion around the limited predictive value of the MAL on AS defects.

6. An alternative approach not discussed is the measurement of MAL in muscle as likely having higher correlations (though technically more difficult).

Reviewer #2: I would like to thank the authors for the possibility to review this manuscript entitled Towards development of a statistical framework to evaluate myotonic dystrophy type 1 mRNA biomarkers in the context of a clinical trial by Kurkiewicz et al. The authors present an interesting approach to evaluate type 1 myotonic dystrophy mRNA biomarkers. They describe how CGT length may be affected by some previously described candidate mRNA by means of machine learning. The topic is relevant and interesting, although, the methodology to quantify treatment efficacy in the context of emerging DM1 treatments, needs some clarifications and could be implemented. Here you will find some comments:

1. Line #124: I would like to underline that the authors mention that the database has 35 patients but, nevertheless, line #127 the patients seem to be 36. It seems that the true sample was 36 but needs clarification as it is confusing.

2. Data preparation needs to be clearer. The authors could clarify which are the dependent and independent variables, as it is not easy to be deducted in the text. Pipeline description is not clear, i.e. what information provides the fact that there are 19,826 files? (#186-#212).

3. Splitting data with only 28/35 patients may cause bias in estimations. (Xu, Y., Goodacre, R., 2018). So, I would like to recommend using all patients and conduct a standard model diagnostic. No model diagnosis has been provided.

4. I would like to suggest the authors to characterize the descriptive analysis provided, including but not limited to MAL variable.

5. R2 is used to evaluate predictive power of each model but this strategy does not penalize by number of predictors. Other methods are suggested as, for instance, AIC/BIC.

6. I think that some clarification could be useful in understand how linear regression was conducted as it seems that there are almost the same number of predictors than patients for DM1-AS. Thus, R2 should be much higher than reported or even not calculable.

7. Relation between previous sections and power analysis section needs to be clearer.

Xu, Y., Goodacre, R. (2018). On Splitting Training and Validation Set: A Comparative Study of Cross-Validation, Bootstrap and Systematic Sampling for Estimating the Generalization performance of Supervised Learning. Journal of Analysis and Testing, 2(3), 249-262.

6. PLOS authors have the option to publish the peer review history of their article (what does this mean?). If published, this will include your full peer review and any attached files.

Reviewer #1: No

Reviewer #2: Yes: JA Carbonell-Asíns

---

## [Author Response · Author response to Decision Letter 0]

13 Jan 2020

Response to reviewers is attached as a word document and labelled Response to Reviewers, per instructions in the decision letter

---

## [Decision Letter · Decision Letter 1]

29 Jan 2020

PONE-D-19-24916R1

Towards development of a statistical framework to evaluate myotonic dystrophy type 1 mRNA biomarkers in the context of a clinical trial

PLOS ONE

Dear Mr. Kurkiewicz,

Thank you for submitting your manuscript to PLOS ONE. After careful consideration, we feel that it has merit but does not fully meet PLOS ONE’s publication criteria as it currently stands. Therefore, we invite you to submit a revised version of the manuscript that addresses the points raised during the review process.

Specifically, the statistician reviewer still finds two relevant technical issues that require a detailed response by the authors

We would appreciate receiving your revised manuscript by Mar 14 2020 11:59PM. To enhance the reproducibility of your results, we recommend that if applicable you deposit your laboratory protocols in protocols.io, where a protocol can be assigned its own identifier (DOI) such that it can be cited independently in the future. For instructions see: http://journals.plos.org/plosone/s/submission-guidelines#loc-laboratory-protocols

We look forward to receiving your revised manuscript.

Kind regards,

Ruben Artero, Ph.D.

Academic Editor

PLOS ONE

Reviewers' comments:

Reviewer's Responses to Questions

**Comments to the Author**

1. If the authors have adequately addressed your comments raised in a previous round of review and you feel that this manuscript is now acceptable for publication, you may indicate that here to bypass the “Comments to the Author” section, enter your conflict of interest statement in the “Confidential to Editor” section, and submit your "Accept" recommendation.

Reviewer #1: All comments have been addressed

Reviewer #2: (No Response)

2. Is the manuscript technically sound, and do the data support the conclusions?

Reviewer #1: Yes

Reviewer #2: Partly

3. Has the statistical analysis been performed appropriately and rigorously? 

Reviewer #1: Yes

Reviewer #2: No

4. Have the authors made all data underlying the findings in their manuscript fully available?

Reviewer #1: Yes

Reviewer #2: Yes

5. Is the manuscript presented in an intelligible fashion and written in standard English?

Reviewer #1: Yes

Reviewer #2: Yes

6. Review Comments to the Author

Reviewer #1: (No Response)

Reviewer #2: I would like to thank the authors for all clarifications in the reviewing of the manuscript entitled “Towards development of a statistical framework to evaluate myotonic dystrophy type 1 mRNA biomarkers in the context of a clinical trial” by Kurkiewicz et al. However I have still some follow-up comments regarding two of my previous remarks.

1. R2 is used to evaluate predictive power of each model but this strategy does not penalize by number of predictors. Other methods are suggested as, for instance, AIC/BIC.

I can now understand the aim of using R2 but still wonder if the relation between predicted and observed is linear. Moreover, beta coefficient for the slope should be equal to one and beta coefficient for intercept equal to 0. I suggest relating only to root-mean-square deviation (RMSD) and its confidence interval for model comparison.

2. Splitting data with only 28/35 patients may cause bias in estimations. (Xu, Y., Goodacre, R., 2018). So, I would like to recommend using all patients and conduct a standard model diagnostic. No model diagnosis has been provided.

I understand the difficulty of increasing the sample size but statistically speaking this is an important limitation of the study as PLS suffers from reduced accuracy with small dataset which is furtherly aggravated when partitioning the dataset. My major concern is the justification that the sample size to evaluate myotonic dystrophy type 1 mRNA biomarkers is big enough.

7. PLOS authors have the option to publish the peer review history of their article (what does this mean?). If published, this will include your full peer review and any attached files.

Reviewer #1: No

Reviewer #2: Yes: Juan Antonio Carbonell-Asins

---

## [Author Response · Author response to Decision Letter 1]

10 Mar 2020

Dear Editor,

Please find attached my response to the reviewers as "Response_to_reviewers_2_Kurkiewicz_DMBDI.pdf"

Adam

---

## [Decision Letter · Decision Letter 2]

16 Mar 2020

Towards development of a statistical framework to evaluate myotonic dystrophy type 1 mRNA biomarkers in the context of a clinical trial

PONE-D-19-24916R2

Dear Dr. Kurkiewicz,

We are pleased to inform you that your manuscript has been judged scientifically suitable for publication and will be formally accepted for publication once it complies with all outstanding technical requirements.

With kind regards,

Ruben Artero, Ph.D.

Academic Editor

PLOS ONE

Additional Editor Comments (optional):

Reviewers' comments:

Reviewer's Responses to Questions

**Comments to the Author**

1. If the authors have adequately addressed your comments raised in a previous round of review and you feel that this manuscript is now acceptable for publication, you may indicate that here to bypass the “Comments to the Author” section, enter your conflict of interest statement in the “Confidential to Editor” section, and submit your "Accept" recommendation.

Reviewer #2: All comments have been addressed

2. Is the manuscript technically sound, and do the data support the conclusions?

Reviewer #2: Yes

3. Has the statistical analysis been performed appropriately and rigorously? 

Reviewer #2: Yes

4. Have the authors made all data underlying the findings in their manuscript fully available?

Reviewer #2: Yes

5. Is the manuscript presented in an intelligible fashion and written in standard English?

Reviewer #2: Yes

6. Review Comments to the Author

Reviewer #2: (No Response)

7. PLOS authors have the option to publish the peer review history of their article (what does this mean?). If published, this will include your full peer review and any attached files.

Reviewer #2: Yes: Juan Antonio Carbonell-Asins

---

## [Editor Report · Acceptance letter]

23 Mar 2020

PONE-D-19-24916R2 

Towards development of a statistical framework to evaluate myotonic dystrophy type 1 mRNA biomarkers in the context of a clinical trial 

Dear Dr. Kurkiewicz:

I am pleased to inform you that your manuscript has been deemed suitable for publication in PLOS ONE. Congratulations! Your manuscript is now with our production department. 

With kind regards,

on behalf of

Dr. Ruben Artero 

Academic Editor

PLOS ONE